# Time series cross-correlation between home range and number of infected people during the COVID-19 pandemic in a suburban city

**Haruka Kato** [ID]*, **Atsushi Takizawa** [ID]

Department of Housing and Environmental Design, Graduate School of Human Life and Ecology, Osaka Metropolitan University, Osaka, Japan

* haruka-kato@omu.ac.jp

## Abstract

Control of human mobility is one of the most effective measures to prevent the spread of coronavirus disease 2019 (COVID-19). However, the imposition of emergency restrictions had significant negative impacts on citizens' daily lives. As vaccination progresses, we need to consider more effective measures to control the spread of the infection. The research question of this study is as follows: Does the control of home range correlate with a reduction in the number of infected people during the COVID-19 pandemic? This study aims to clarify the correlation between home range and the number of people infected with SARS-CoV-2 during the COVID-19 pandemic in Ibaraki City. Home ranges are analyzed by the Minimum Convex Polygon method using mobile phone GPS location history data. We analyzed the time series cross-correlation between home range lengths and the number of infected people. Results reveal a slight positive correlation between home range and the number of infected people after one week during the COVID-19 pandemic. Regarding home range length, the cross-correlation coefficient is 0.4030 even at a lag level of six weeks, which has the most significant coefficient. Thus, a decrease in the home range is a weak factor correlated with a reduction in the number of infected people. This study makes a significant contribution to the literature by evaluating key public health challenges from the perspective of controlling the spread of the COVID-19 infectuion. Its findings has implications for policy makers, practitioners, and urban scientists seeking to promote urban sustainability.

## Introduction

### Background

The coronavirus disease 2019 (COVID-19) pandemic has drastically changed our daily lives. The rapid increase in the number of infected people risks causing a breakdown of the medical system. Control of human mobility is considered one of the most effective measures to prevent the rapid spread of COVID-19 [1]. For example, in the Osaka metropolitan area, states of emergency have been declared four times since January 2020 [2], with more substantial restrictions imposed on the activities of people living in areas closer to the city center [3]. The

**Data Availability Statement:** The data of the number of people infected with SARS-CoV-2 is available from references of website of Ibaraki City Government. The link of the website is the following (in Japanese): https://www.city.ibaraki.

osaka.jp/kikou/kenkoi/kenkou/menu/2covid19.
html. On the other hand, the GPS Location History
data (LH data) cannot be shared publicly because
of mobile phone users' confidentiality and the legal
restrictions with authors and Agoop corporation.
However, researchers can use the LH data, if they
contract with the Agoop Corpolation Institutional
Data Access (contact via ESRI Japan) for
researchers who meet the criteria for access to
confidential data. The contact information link is
the following E-mail address: gisinfo@esrij.com
(ESRI Japan, Customer Support Service)

**Funding:** Funded studies H.K., 21K14318, JSPS
KAKENHI, https://kaken.nii.ac.jp/en/grant/
KAKENHI-PROJECT-21K14318/ H.K., 220201, Dai-
ichi Life Foundation, http://group.dai-ichi-life.co.jp/
d-housing/boshu.html T.A., 2102C, International
Association of Traffic and Safety Sciences, https://
www.iatss.or.jp/en/ The funders had no role in
study design, data collection and analysis, decision
to publish, or preparation of the manuscript.

**Competing interests:** The authors have declared
that no competing interests exist.

Subcommittee on Novel Coronavirus Disease Control in Japan requested citizens to reduce human mobility by 50% during this time [4]. However, the imposition of emergency restrictions had significant negative impacts on the daily lives of citizens. For example, excessive restrictions caused a deterioration of mental health [5]. As vaccination progresses, we need to consider more effective measures to control the spread of the infection.

The research question of this study is as follows: Does the control of home range correlate with a reduction in the number of infected people during the COVID-19 pandemic? In other words, this study verifies the possibility of predicting the number of infected people based on the control of human mobility. In particular, it is difficult to predict the number of infected people in suburban cities because of their high human mobility. Therefore, this study focuses on the aspect of home range in human mobility. Home range is defined as the areas that individuals traverse in the course of their daily activities, such as working and shopping [6]. Home range is an essential indicator for policymakers to assess and facilitate the achievement of the daily mobility activities of residents. For example, the "Location Optimization Plan" [7] is a policy that aims to maintain the home range around transit stations in Japan. Understanding the time series cross-correlation between home range and the number of infected people would support policymakers to develop policies for controlling the spread of the COVID-19 infection.

## Purpose

This study aims to clarify the time series cross-correlation between home range and the number of people infected with SARS-CoV-2 during the COVID-19 pandemic in a suburban city. Home ranges were analyzed by the Minimum Convex Polygon (MCP) method using mobile phone GPS location history (LH) data. LH data includes location history data collected from individual devices, unlike area-based data such as Google mobility reports [8].

In this study, the pandemic period was set from January 2020 to July 2021. In Japan, the first case of SARS-CoV-2 infection was confirmed in January 2020. From April to May 2020, the first state of emergency was declared for all prefectures. In 2021, states of emergency were declared repeatedly, mainly in Osaka prefectures, from January to February, April to June. After July 2021, vaccination progress in Japan. Therefore, the analysis period, from January 2020 to July 2021, is thought to be the COVID-19 pandemic period in Japan. A timeseries analysis was conducted using panel data of every Wednesday from April 2020 to July 2021, a time frame where four waves of the pandemic were witnessed in Japan. This study analyzed the time series cross-correlation between the home range lengths and the number of infected people.

The case study research was conducted in Ibaraki City, which is a typical suburban city in the Osaka metropolitan area. Fig 1 shows the location of Ibaraki City, which has a population of approximately 280,000, and an area of 10 km east-west and 17 km north-south [9]. The map in Fig 1 was the open-source Arc GIS PRO and complied with the copyright [10]. Due to the city's extensive train network, residents can commute in about 30 min to Osaka City or Kyoto City. Fig 1 shows the distance from the central area of Ibaraki City, where the Ibaraki City Government Hall is located. Osaka Station, located in the central area of Osaka, is 15 km away from the central area of Ibaraki City. This distance helps us understand the home range extent.

In Ibaraki City, home range decreased by approximately 50% during the first state of emergency [11]. In addition, dense clusters of people were formed in the parks as well as in the stations during the period [12]. The frequency of walking and bicycle trips increased during the period [13]. These studies suggest that the home range had decreased to a neighborhood scale due to the restrictions imposed during the COVID-19 pandemic in Ibaraki City.

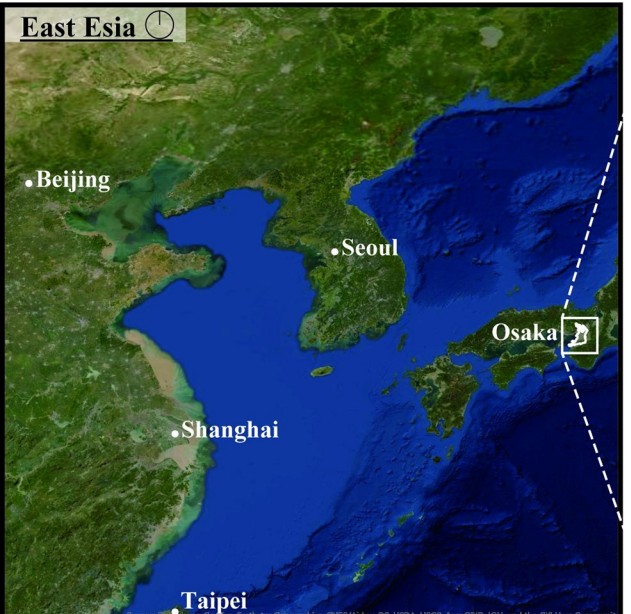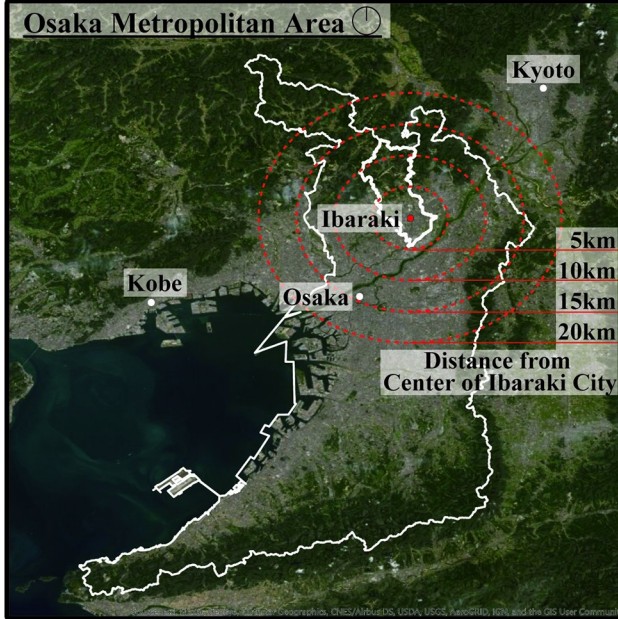

**Fig 1. Locations of Osaka Prefecture and Ibaraki City.** The thin lines indicate the location of Osaka Prefecture, and the thick lines indicate the location of Ibaraki City. The red point is the central area of Ibaraki City, where the Ibaraki City Government Hall is located. The red dotted circles indicate the distance from the center of Ibaraki City. The distances are 5 km, 10 km, 15 km, and 20 km.

## Literature review

Many studies analyzed the factors associated with the number of people infected with SARS--CoV-2. For example, it was reported that the risk of infections was correlated with chronic exposure to various outdoor air pollution [14]. One effective policy for redusing the airpolusion is to reduce the human mobility related to the transportation infrastructure [15,16]. During the lockdown restricting human mobility, it was found to reduce air pollution in some cities [17,18]. Therefore, the originality of this study is to focus on human mobility to estimate the number of people infected with SARS-CoV-2.

For the analysis of human mobility, many studies used mobile phone data. Regarding mobile phone data, GPS location history data is the state-of-the-art technology that has started to gain attention in the COVID-19 pandemic. In particular, many studies focused on Wuhan, where the first infected person was identified. In the early stages of the pandemic, the spatial distribution of the number of infected people was found to explain the movement of the population from Wuhan between January 1 and January 24, 2020 [19]. After January 24, 2020, findings suggested that the lockdown reduced the number of COVID-19 cases in Wuhan by limiting human mobility within the city [20]. Based on these results, simulations to determine the feasibility of measures for the successful control and containment of the COVID-19 pandemic showed the necessity of restricting human mobility by 20%-40%, using the case of Shenzhen in China [21]. In the United States, human mobility was severely restricted during the early stages of the pandemic [22]. Using mobile phone data across the United States from January 1 to April 20, 2020, it was found that COVID-19 transmission correlated strongly with mobility patterns [23]. In addition, based on mobile phone data from March 1 to June 9, 2020, a positive correlation was found between the number of infected people and mobility inflow at the country level in the United States [24]. Compared to these studies, the originality of this

study is to clarify the time series cross-correlation between human mobility and the number of people infected with SARS-CoV-2.

Regarding the time series cross-correlation, it was found that the restriction of human mobility has a time lag effect on the growth rates of COVID-19 cases [25]. In China, human mobility is strongly correlated with COVID-19 cases, with lags of 10 days and a correlation coefficient of 0.68 from January 17 to February 19, 2020 [26]. In Indonesia, human mobility is also correlated with COVID-19 cases with lags of 7–14 days and a correlation coefficient of about 0.50 from March 1 to July 31, 2020 [27]. In the European context, the spread of COVID-19 was positively correlated with the number of people staying in each area and with human mobility between March 1 to June 6, 2020, at lag levels of one, two, and three weeks [28]. This could be attributed to the fact that in Europe, lockdowns generally affect long-distance travel behavior [28]. In addition, between January 1 and April 15, 2020, it was found that the estimated adequate reproduction number of COVID-19 correlated strongly with human mobility (or social contact) in Tokyo, Japan [29]. The changes in human mobility pertaining to nightlife spaces were more significantly associated with the number of COVID-19 cases [30]. Those studies indicate that the number of infected people correlated with human mobility during the early stages of the pandemic.

The novelty of this study is to analyze the time series cross-correlation during the pandemic, which is from every Wednesday from April 2020 to July 2021. In Japan, the state of emergency was called a "soft lockdown" [31] because the Japanese government did not restrict the activities of individuals [32]. Therefore, most citizens could at least go out in a limited capacity even under the state of emergency. That suggests that home range might correlate slightly with the spread of the infection. This study will allow policymakers to develop policies for controlling the infection during the COVID-19 pandemic in the medium to long term.

## Materials and methods

### Location history data

The study used LH data collected by Agoop Corporation. Agoop Corporation collected the LH data by obtaining consent from users, who contracted with a specific mobile phone carrier company or installed specific applications [33]. All participants were provided with information regarding the type of data collected, purpose of use and provision to third parties, and a privacy policy [34]. That means that informed consent was obtained from all subjects based on the privacy policy. However, the consent is not written paper because it is digital data using mobile phone. Additionally, the subjects can stop sending their human mobility data anytime by changing their mobile phones' settings. Agoop Corporation provides anonymized LH data for research purposes. Due to the availability of high-quality data in a Japanese context, many studies utilized LH data relating to the spread of COVID-19 [10–12]. This research protocol was approved by the ethics committee of the Graduate School of Life Science, Osaka City University (No.21-40). Additionally, all methods were carried out in accordance with "Guidelines for the Use of Device Location Data," a common regulation for location data analysis in Japan [35]. The guideline prohibits using GPS data for any purpose that involves identifying individual users to protect the privacy of users' GPS location history.

Depending on the mobile phone type, the LH data were collected in the form of logs approximately every 15 min, and the LH data were obtained from mobile phones with users' consent. In Ibaraki City, the number of logs was approximately 1,600,000 per day, and the number of users was approximately 12,000, indicating that 5% of the residents of Ibaraki City was adequate for analyzing the home range of residents.

The variables in the LH data used in this study were user ID, year, month, day, hour, minutes, and latitude and longitude. The user IDs are anonymized 96-digit alphanumeric codes, the permanent ID assigned to each device, and enable panel data analysis.

## Time series cross-correlation

This study used a time series analysis method to analyze the cross-correlation between home range length and the number of infected people. The time-series cross-correlation allows us to understand the similarity of data in a time series and the lag of the period. In this study, the lag was set to eight weeks, considering that the duration between the date of exposure and onset of symptoms is usually a few weeks [36].

The analysis period was from April 2020 to July 2021, during which Japan experienced four waves, and a state of emergency was imposed four times in Ibaraki City. On April 7, 2020, when a state of emergency was declared in all prefectures, the Japanese government requested people to stay at home. Following the end of this state of emergency, the Japanese government attempted to recover the economy through various measures such as the "Go-To Travel" campaign for the hotel and restaurant industries [37]. However, by the winter of 2020, the number of infected people had gradually increased [38]. A second state of emergency was subsequently declared from January 14 to February 28, 2021. The Japanese government also developed priority preventive measures before the next emergency declaration [2]. These priority preventive measures were in effect from April 5 to June 24, 2021, and a third state of emergency was declared from April 25 to June 20, 2021. A second set of priority preventive measures was introduced from June 21 to August 1, 2021, and a fourth state of emergency was declared from August 2 to September 30, 2021. While vaccination for healthcare workers began in February 2021 [2], vaccination for older people and adults in Ibaraki City began in May 2021. This study focuses on the period between April 2020 and July 2021, prior to the fourth emergency declaration, shortly after the beginning of the vaccination campaign.

## Home range length

This study followed the method of the authors' previous study [11]. Home range length was analyzed using the MCP method, which analyzes zones that connect the outermost observation points [39]. The advantage of this method is its simplicity and intuitive use.

Using the MCP method, this study analyzed two types of home range lengths (HR-lengths): HR-length (Farthest Distance) and HR-length (Total Travel Distance). HR-length (Farthest Distance) is the distance to the farthest point moved from the home. The home was estimated by the starting home point after 0:00. The HR-length (Farthest Distance) allows us to understand the extent of the area traveled. HR-length (Total Travel Distance) is the total distance covered every day in the time period under consideration. Since the data are acquired every 15 min, the actual HR-length (Total Travel Distance) cannot be calculated. However, the HR-length (Total Travel Distance) allows us to estimate the total travel distance. The HR-length (Total Travel Distance) was used as an indicator in a previous study [29].

The analysis process is summarized in Fig 2. The map in Fig 2 was the open-source Arc GIS PRO and complied with the copyright [10]. Changes in HR-length (Farthest Distance) and HR-length (Total Travel Distance) during the COVID-19 pandemic were analyzed. In addition, these two HR-lengths were analyzed the cross-correlation with the number of people infected.

The study analyzed LH data in Ibaraki City. However, this data included the logs of people who only passed through Ibaraki City, such as people commuting from Tokyo to Osaka by express. Therefore, to isolate the data of people living in Ibaraki City, the study extracted user

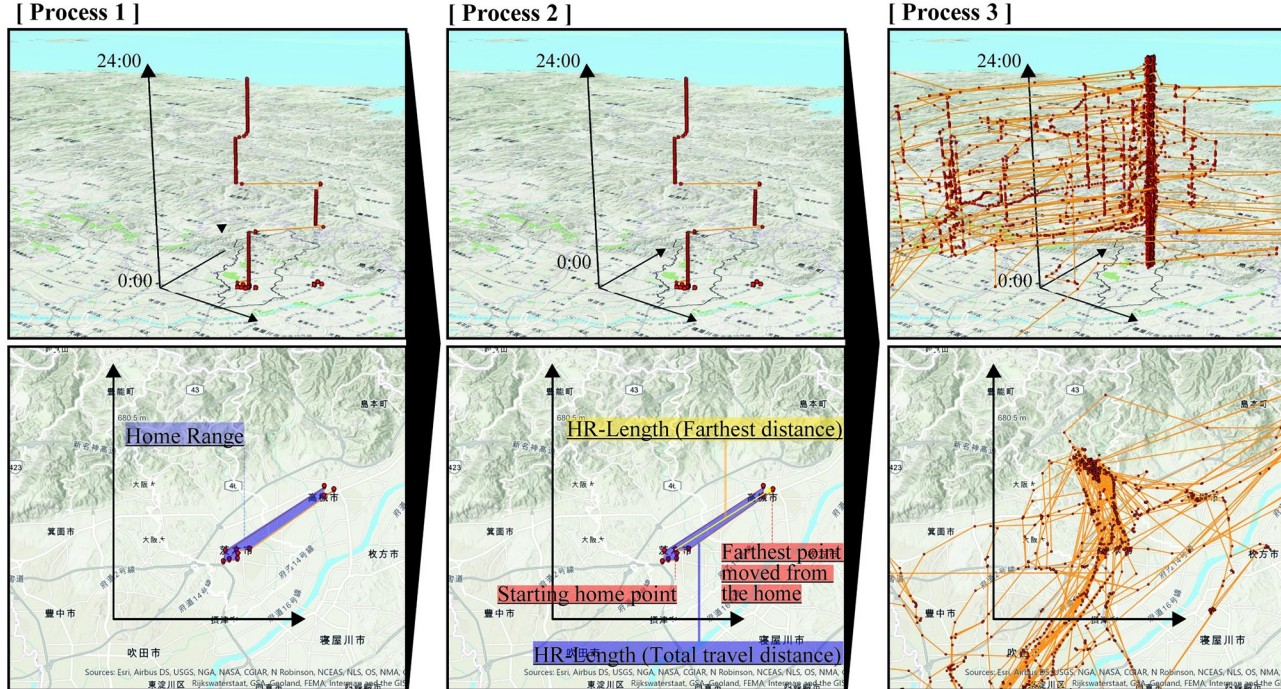

**Fig 2. Analysis process to analyze the home range.** The upper figure is a 3D spatiotemporal map, and the lower figure is 2D. In process 1, the personal space-time paths between 0:00 to 23:59 h were drawn. In process 2, the home range distance for each mobile phone user is calculated from the latitude and longitude difference between the starting home point and the farthest point moved from the home after 0:00 h. The home was estimated by the starting home point after 0:00. In process 3, the distance of the mobile phone user is averaged to calculate the HR-length (Farthest Distance) and HR-length (Total Travel Distance) of every day. The space-time paths are the imaginary paths of 50 people.

IDs of the first log located in Ibaraki City after 0:00 h of every day. The study then analyzed the user ID data that appeared for more than two days in the analyzed period. The intent of the study was to analyze the user IDs of individuals living in Ibaraki City, not those just passing through it. With respect to the LH data of the user ID, this study analyzed the home range from 0:00 to 23:59 h.

This study analyzed the change in home range based on the LH data from every Wednesday, considering the home range changes that occurred between weekdays and holidays [11]. On weekdays, people are involved in steady-state activities such as work. Wednesday was chosen because suitable for analyzing the impact of the COVID-19 pandemic. Wednesday was chosen as the most appropriate day of the week to analyze the impact of the COVID-19 pandemic for a number of reasons. Wednesday is the weekday with the fewest holidays from April 2020 to July 2020: only May 6, 2020, and May 5, 2021, were holidays. Furthermore, December 30, 2020, marked the beginning of the year-end vacation. Moreover, on December 30, many people tend to be off work and school during the year-end vacations.

## Number of infected people

This study analyzed the number of people infected with SARS-CoV-2 in Ibaraki City. Data published on the webpage of the Ibaraki City Government was used for the purpose of the study [40]. The number of people infected was calculated as the total number of people infected each week from Monday to Sunday. Since many hospitals are closed on Sundays, this calculation method was deemed appropriate for use in the Japanese context.

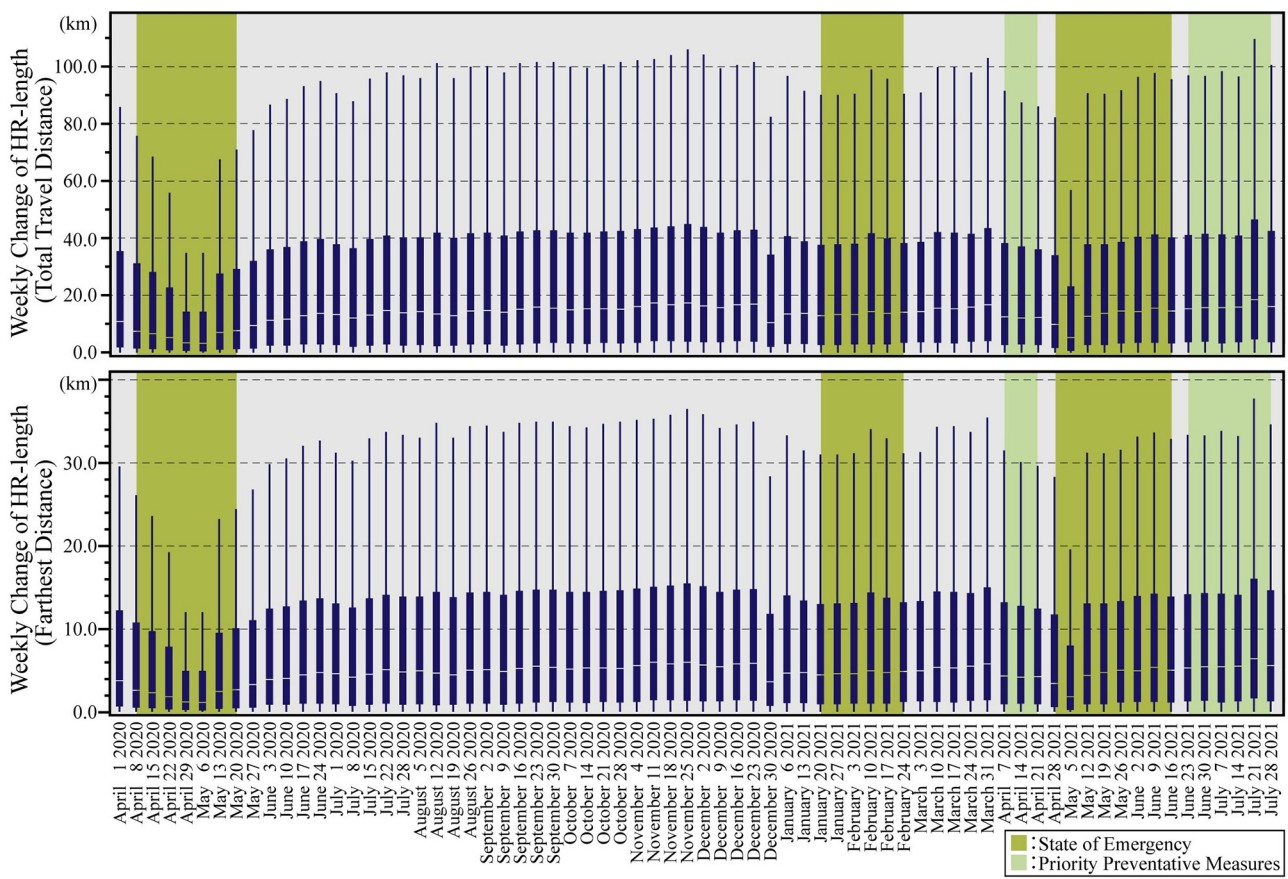

**Fig 3. Weekly change of HR-length.** The upper graph shows the weekly changes in HR-length (Farthest Distance). The lower graph shows the weekly changes in HR-length (Total Travel Distance). Green periods are during the state of emergency, and yellow-green periods are during the priority prevention measures. The box-plot diagrams do not depict the outliers.

## Results

### Change of home range length

Fig 3 shows the box-plot diagrams of weekly changes in HR-length (Farthest Distance) and HR-length (Total Travel Distance) from April 2020 to July 2021; Fig 4 shows the bar-graphs of monthly changes in HR-length (Farthest Distance) and HR-length (Total Travel Distance) from April 2020 to July 2021. In both figures, weekly and monthly data followed similar trends. In particular, the home range showed a significant decrease every week from April to May 2020, when the first state of emergency was declared. This result was verified by previous research [11–13]. However, since June 2020, the home range has gradually increased. It was found that the HR-length decreased significantly only from April to May 2020, when the first emergency declaration was issued, but thereafter, travel to Ibaraki City and Osaka City did not change significantly.

Figs 3 and 4 depict that there are no significant changes except on December 31, 2020, and May 5, 2021, which were national holidays, and July 21, 2021, right before the Tokyo Olympics 2020 were held. The home range was found to have decreased during those periods when priority preventive measures were in effect, while the home range tended to increase during emergency periods. The results also suggest that citizens had changed their mobility behavior before the government declared a state of emergency.

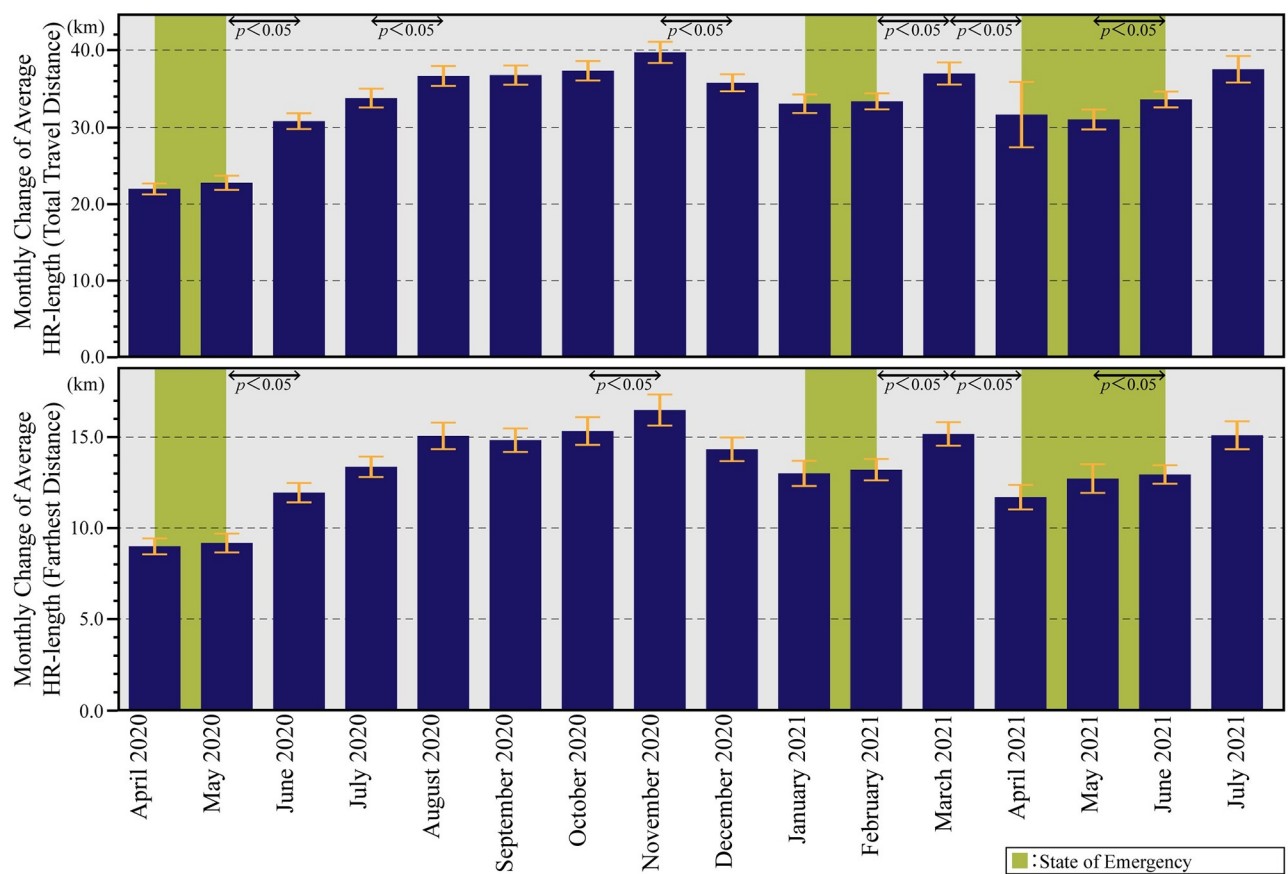

**Fig 4. Monthly Change of HR-length.** The upper graph shows the monthly changes in HR-length (Farthest Distance). The lower graph shows the HR-length (Total Travel Distance). The figures shows the average and 95% intervals of the HR-length in a time series. Besides, the Wilcoxon rank-sum test indicates significant differences in the average values for each month. Green periods are during the state of emergency, and yellow-green periods are during the priority prevention measures.

## Change in the number of infected people

Fig 5 illustrates a bar graph of weekly changes in the number of people infected with SARS-CoV-2 in Ibaraki City between March 29, 2020, and July 31, 2021. Fig 5 shows that Ibaraki City has experienced four increases and decreases in the number of infected people. The first wave lasted from April to May 2020, the second from July to September 2020, the third from December 2020 to February 2021, and the fourth from March to June 2021. The fifth wave began in July 2021. Each wave lasted progressively longer than the preceding one.

States of emergency were declared for the first, third, and fourth waves. Following each declaration of emergency, the number of infected people decreased significantly. In particular, the number of infected people decreased to zero from May 17–23, 2020, at which time the first emergency declaration was lifted; to five people from February 21–27, 2021, when the second emergency declaration was lifted; and to 13 people from June 13–19, 2021, when the third emergency declaration was lifted. This suggests that the declaration of a state of emergency effectively led to a decrease in the number of infected people.

## Cross-correlation of home range length and infected people

Fig 6 presents the time series cross-correlation of home range length and number of infected people. It was found that the number of infected people was slightly more correlated with HR-

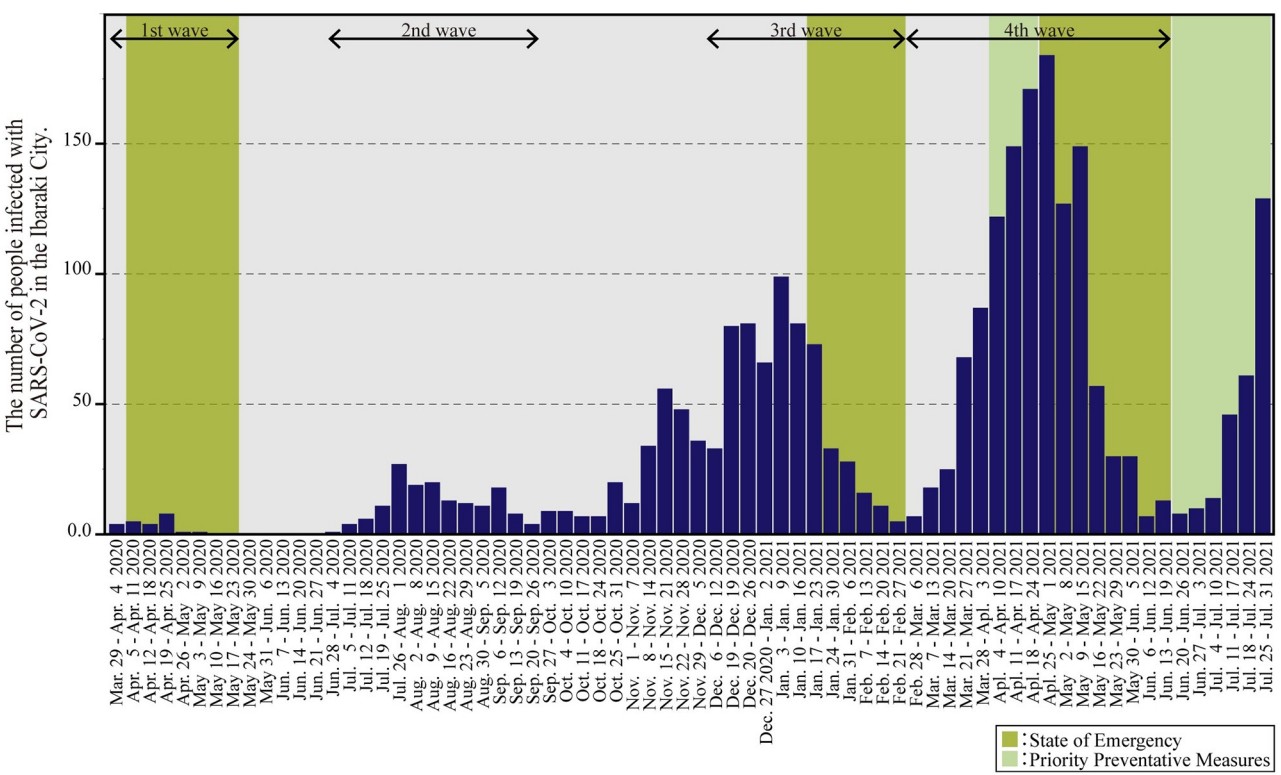

**Fig 5. Weekly change of the number of infected people between March 29, 2020, and July 31, 2021.** Osaka Prefecture experienced four waves during this time. Green periods are during the state of emergency, and yellow-green periods are during the priority prevention measures.

| Lag | CCC Total Travel Distance | -.8 -.6 -.4 -.2 0 .2 .4 .6 .8 | CCC Farthest Distance | -.8 -.6 -.4 -.2 0 .2 .4 .6 .8 |
|---|---|---|---|---|
| -8 | -0.0568 | | -0.0951 | |
| -7 | -0.0878 | | -0.1346 | |
| -6 | -0.1207 | | -0.1518 | |
| -5 | -0.1317 | | -0.1692 | |
| -4 | -0.1192 | | -0.1703 | |
| -3 | -0.1010 | | -0.1231 | |
| -2 | -0.0494 | | -0.0628 | |
| -1 | 0.0329 | | 0.0342 | |
| 0 | 0.1546 | | 0.1345 | |
| 1 | 0.2398 | | 0.2212 | |
| 2 | 0.2974 | | 0.2805 | |
| 3 | 0.3447 | | 0.2897 | |
| 4 | 0.3649 | | 0.3525 | |
| 5 | 0.3744 | | 0.3475 | |
| 6 | 0.4030 | | 0.3950 | |
| 7 | 0.3543 | | 0.3314 | |
| 8 | 0.3016 | | 0.2611 | |

**Fig 6. Cross-correlation of home range lengths and the number of infected people.** CCC Total Travel Distance is HR-length (Total Travel Distance). CCC Farthest Distance is HR-length (Farthest Distance). In cross-correlation, the lag was set to eight weeks.

length (Total Travel Distance) than HR-length (Farthest Distance). In addition, over the lag level of zero weeks, HR-length and the number of infected people are positively correlated. The results indicate that a decrease in HR-length leads to a decline in the number of infected people. However, there was only a slightly positive cross-correlation between HR-length and the number of infected people. Specifically, the cross-correlation coefficient (CCC) of HR-length (Total Travel Distance) (CCC $_{Total\ Travel\ Distance}$) is 0.1546 at the lag level of zero weeks. As the lag increases, CCC $_{Total\ Travel\ Distance}$ also increases, and CCC $_{Total\ Travel\ Distance}$ exceeds 0.2 after a lag level of one week. However, even at the lag level of six weeks, the largest CCC, the CCC $_{Total\ Travel\ Distance}$ is 0.4030. Similarly, the cross-correlation coefficient of HR-length (Farthest Distance) (CCC $_{Farthest\ Distance}$) is 0.1345 at the lag level of zero weeks, and 0.3950 at the lag level of six weeks, which has the largest CCC.

To summarize these results, home range was found to be slightly positively correlated with the number of infected people after six weeks, with the highest correlation coefficient being 0.40. This means that changing the home range correlates with a decrease in the number of infected people, but it is not a strong correlation.

## Discussion and conclusion

In conclusion, this study clarifies a slight positive correlation between home range and the number of infected people after one week during the COVID-19 pandemic in Ibaraki City. The positive correlations validated the results of previous studies [24,28]. In addition, prior research that analyzed the early-stage period clarified a strong correlation [23,29]. The result of this study was significant because it clarifies the slight positive correlation between home range and the number of infected people in the COVID-19 pandemic.

It was found that the number of infected people was slightly more correlated with HR-length (Total Travel Distance) than HR-length (Farthest Distance). Controlling travel distance is more effective than controlling the tendency to go out farther. With regard to HR-length (Total Travel Distance), the CCC was found to be 0.4030 even at a lag level of six weeks, which had the most significant coefficient. The result differed from previous studies that clarified the strong coefficient during the early stage of the pandemic [25–27]. Besides, it was inexplicable at a lag level of six weeks. The reason for this is the length of the incubation period of SARS--CoV-2; it takes approximately one–two weeks from the date of infection for the first symptoms to appear [36]. Specifically, after the emergency declaration was issued, the number of infected people decreased significantly in approximately one–two weeks. However, the home range had decreased even before the emergency declaration was issued, and the decrease was not significant. This might be the reason for the low correlation coefficient. That result was also found that the COVID-19 cases were slightly compensated for positive indirect effects through human mobility [41]. The novelty of the present study is that decrease in the home range is a weak factor correlated with a reduction in the number of infected people.

The conclusions suggest that we need to take some measures other than restricting the home range for decreasing in the number of infected people; for instance, restrictions imposed under the state of emergency in the Osaka Prefecture, which applied to the residents of Ibaraki City during this period. Citizens were required to refrain from not only non-urgent outings but also from drinking alcohol in a group on the street or in a park; the operation of restaurants after 8:00 p.m. was also suspended, and large-scale events were prohibited [42]. Restrictions with regard to wearing face masks might also have had an impact on the rate of infection. For example, as the home range expands, one factor influencing the spread of infection might be an increase in the number of situations in which people remove their masks, such as while eating lunch and smoking [43]. In addition, during the third emergency declaration,

household infections had become a significant problem, which was associated with sharing a bedroom and speaking with an index case individual for 30 min or longer [44].

Based on these results, it is possible to improve current measures for an emergency declaration. For instance, instead of controlling human mobility, the number of infected people could be effectively reduced by the imposition of mask mandates, reducing the opening hours of restaurants, and increasing the use of hotel facilities for medical treatment. These measures would also make it possible for people to fulfill their work and study commitments while taking steps to protect themselves from infection.

The limitation of this study is that we analyzed only the indicator of home range in Ibaraki City. It is necessary to analyze not only the distance traveled using LH data, but also the place of stay using area-based data. The analysis might provide a higher correlation coefficient. Further, due to privacy issues, the discrepancies between samples of the number of infected people and human mobility pose a research challenge. Therefore, it is also necessary to analyze the actual number of infected people and the distance they travel. Moreover, we analyzed data from Ibaraki City, a suburban city, but human mobility in central cities should also be considered in determining the significant factors influencing infection spread. In the future, it is necessary to study different types of cities to examine the correlation between home range and the number of infected people in a Japanese context, such as Osaka City, the more metropolitan capital of the Osaka Prefecture. In addition, for urban sustainability, it is essential to study the other factors that correlate with the number of infected cases, such as air pollution, using existing and upcoming biomass measurement missions.

## Author Contributions

**Conceptualization:** Haruka Kato.

**Data curation:** Haruka Kato.

**Formal analysis:** Haruka Kato.

**Funding acquisition:** Haruka Kato.

**Investigation:** Haruka Kato.

**Methodology:** Haruka Kato.

**Project administration:** Haruka Kato.

**Resources:** Haruka Kato.

**Software:** Haruka Kato.

**Supervision:** Haruka Kato.

**Validation:** Haruka Kato, Atsushi Takizawa.

**Visualization:** Haruka Kato.

**Writing – original draft:** Haruka Kato.

**Writing – review & editing:** Haruka Kato, Atsushi Takizawa.

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
