## [Decision Letter · Decision Letter 0]

28 Jun 2022

PONE-D-22-09773Time series cross-correlation between home range and number of infected people during the medium term of COVID-19 Pandemic in a suburban cityPLOS ONE

Dear Dr. KATO,

Thank you for submitting your manuscript to PLOS ONE. After careful consideration, we feel that it has merit but does not fully meet PLOS ONE’s publication criteria as it currently stands. Therefore, we invite you to submit a revised version of the manuscript that addresses the points raised during the review process.

We look forward to receiving your revised manuscript.

Kind regards,

Ghaffar Ali, PhD

Academic Editor

PLOS ONE

Journal Requirements:

“NO”

5. We note that Figures 1 and 2 in your submission contain map images which may be copyrighted. All PLOS content is published under the Creative Commons Attribution License (CC BY 4.0), which means that the manuscript, images, and Supporting Information files will be freely available online, and any third party is permitted to access, download, copy, distribute, and use these materials in any way, even commercially, with proper attribution. For these reasons, we cannot publish previously copyrighted maps or satellite images created using proprietary data, such as Google software (Google Maps, Street View, and Earth). For more information, see our copyright guidelines: http://journals.plos.org/plosone/s/licenses-and-copyright.

   a. You may seek permission from the original copyright holder of to Figures 1 and 2 publish the content specifically under the CC BY 4.0 license. 

Reviewers' comments:

Reviewer's Responses to Questions

**Comments to the Author**

1. Is the manuscript technically sound, and do the data support the conclusions?

Reviewer #1: Yes

Reviewer #2: Yes

2. Has the statistical analysis been performed appropriately and rigorously? 

Reviewer #1: Yes

Reviewer #2: Yes

3. Have the authors made all data underlying the findings in their manuscript fully available?

Reviewer #1: Yes

Reviewer #2: Yes

4. Is the manuscript presented in an intelligible fashion and written in standard English?

Reviewer #1: Yes

Reviewer #2: Yes

5. Review Comments to the Author

Reviewer #1: This study titled "Time series cross-correlation between home range and number of infected people during the medium term of COVID-19 Pandemic in a suburban city", is an interesting study where the author attempt to investigate the link between home range and COVID-19 infections. My comments are as below:

1) There are a lot of studies published on this topic in recent times, the author needs to explain how this study is different from others.

2) Discussion section should be in more detail, and the author needs to compare the findings with other studies.

Reviewer #2: This manuscript unveils the time series cross-correlation between home range and number of infected people during the medium-term of the COVID-19 pandemic in a suburban city. The article is in good shape and could be accepted for possible publication after addressing the comments given below:

Title:

The “medium-term” may not be suitable in the title, which may put reader to guess about the term.

Abstract: Could have been written better. Also include and elaborate the approach. The first sentence should be broken into 2-3 clear sentences. SBM – write in full when appeared first in the abstract and introduction.

Introduction:

The introduction is very good; the authors demonstrate a thorough knowledge of the published literature and highlight the importance and background to carry out this investigation. Generally, it is reasonable, to follow the current pattern, which is more clear, but it might be good to follow the stat of the art format for the introduction. Then, elaborate the objectives with focus before jumping on the description of sessions.

Also include some regional and local studies on the impact of covid-19 on mobility and its associated impact other environmental indicators for instance air pollution, it has been noted the different mobility patterns and measures across the Asia. Some references are given below which could be considered.

Figure 1 should be move in the Location History Data.

Materials and Methods

Methods are technically strong and well explained.

Figures’ captions are good and self-explanatory.

Results:

Results are described in detail.

Line 177 Section 3.1 in not mentioned in the manuscript – probable formatting issues.

Figures’ captions are good and self-explanatory.

Discussion

Authors have combined Discussion with Conclusion which is not a usual practice but could be acceptable. It would be good to add relevance and prospective of the current study for wider area coverage using existing and upcoming biomass measurement missions.

Conclusion

No Comments.

Figure 1 – Please add a proper study area map with location of Japan and the study area in index maps.

 "Changes in air pollution levels after COVID-19 outbreak in Korea." Science of the Total Environment 750 (2021): 141521.

 Associations of air pollution concentrations and energy production dynamics in Pakistan during lockdown. Environ Sci Pollut Res (2022). https://doi.org/10.1007/s11356-021-18071-4

 Environmental Impacts of Shifts in Energy, Emissions, and Urban Heat Island during the COVID-19 Lockdown Across Pakistan. J. Clean. Prod. 125806. https://doi.org/10.1016/j.jclepro.2021.125806

 "Positive association between outdoor air pollution and the incidence and severity of COVID-19. A review of the recent scientific evidences." Environmental Research (2021): 111930.

 Environmental Spatial Heterogeneity of the Impacts of COVI-19 on the Top-20 Metropolitan Cities of Asia-Pacific. Nature Scientific Reports, 11, 20339.  2021

6. PLOS authors have the option to publish the peer review history of their article (what does this mean?). If published, this will include your full peer review and any attached files.

Reviewer #1: No

Reviewer #2: No

---

## [Author Response · Author response to Decision Letter 0]

27 Jul 2022

Dear Reviewers and Editor

We appreciate the time and effort you have dedicated to providing insightful feedback on strengthening our manuscript. Thus, it is with great pleasure that we re-submit our article for further consideration. We have incorporated changes that reflect the detailed suggestions you have graciously provided. We also hope that our edits and the responses we provide below satisfy the issue and concern you have noted. We want you to check the revised manuscript and response letter.

---

## [Decision Letter · Decision Letter 1]

12 Aug 2022

Time series cross-correlation between home range and number of infected people during the COVID-19 pandemic in a suburban city

PONE-D-22-09773R1

Dear Dr. KATO,

We’re pleased to inform you that your manuscript has been judged scientifically suitable for publication and will be formally accepted for publication once it meets all outstanding technical requirements.

Kind regards,

Ghaffar Ali, PhD

Academic Editor

PLOS ONE

Additional Editor Comments (optional):

Reviewers' comments:

Reviewer's Responses to Questions

**Comments to the Author**

1. If the authors have adequately addressed your comments raised in a previous round of review and you feel that this manuscript is now acceptable for publication, you may indicate that here to bypass the “Comments to the Author” section, enter your conflict of interest statement in the “Confidential to Editor” section, and submit your "Accept" recommendation.

Reviewer #1: All comments have been addressed

Reviewer #2: All comments have been addressed

2. Is the manuscript technically sound, and do the data support the conclusions?

Reviewer #1: Yes

Reviewer #2: Yes

3. Has the statistical analysis been performed appropriately and rigorously? 

Reviewer #1: Yes

Reviewer #2: Yes

4. Have the authors made all data underlying the findings in their manuscript fully available?

Reviewer #1: (No Response)

Reviewer #2: Yes

5. Is the manuscript presented in an intelligible fashion and written in standard English?

Reviewer #1: Yes

Reviewer #2: Yes

6. Review Comments to the Author

Reviewer #1: (No Response)

Reviewer #2: I am happy with the revised version of the manuscript. The authors have addressed all the comments and suggestions.

7. PLOS authors have the option to publish the peer review history of their article (what does this mean?). If published, this will include your full peer review and any attached files.

Reviewer #1: **Yes: **Sulaman Muhammad

Reviewer #2: No

---

## [Editor Report · Acceptance letter]

22 Aug 2022

PONE-D-22-09773R1 

Time series cross-correlation between home range and number of infected people during the COVID-19 pandemic in a suburban city 

Dear Dr. Kato:

I'm pleased to inform you that your manuscript has been deemed suitable for publication in PLOS ONE. Congratulations! Your manuscript is now with our production department. 

Kind regards, 

on behalf of

Prof. Ghaffar Ali 

Academic Editor

PLOS ONE